# The Screen Project: Guided Direct-To-Consumer Genetic Testing for Breast Cancer Susceptibility in Canada

**DOI:** 10.3390/cancers13081894

**Published:** 2021-04-15

**Authors:** Steven A. Narod, Nicole Gojska, Ping Sun, Angelina Tryon, Joanne Kotsopoulos, Kelly Metcalfe, Mohammad R. Akbari

**Affiliations:** 1Women’s College Research Institute, Women’s College Hospital, Toronto, ON M5S 1B2, Canada; nicole.gojska@wchospital.ca (N.G.); Ping.Sun@wchospital.ca (P.S.); angelina.tryon@wchospital.ca (A.T.); joanne.kotsopoulos@wchospital.ca (J.K.); kelly.metcalfe@utoronto.ca (K.M.); 2Dalla Lana School of Public Health, University of Toronto, Toronto, ON M5T 3M7, Canada; 3Bloomberg Faculty of Nursing, University of Toronto, Toronto, ON M5T 1P8, Canada

**Keywords:** BRCA1, BRCA2, genetic testing, genetic counselling

## Abstract

**Simple Summary:**

The Screen Project was initiated in 2017 to offer BRCA1 and BRCA2 genetic screening to all Canadians over the age of 18 who wish to know their mutation status. Study subjects registered online and were sent a saliva sample kit. Of the 1269 tested individuals between March 2017 and January 2019, 30 (2.4%) had a pathogenic mutation in BRCA1 or BRCA2 (20 women and 10 men). Seventy-five percent of the female mutation carriers underwent a bilateral mastectomy and/or salpingo-oophorectomy within a year of receiving a positive result. The study results endorse a model where genetic testing is offered directly to the consumer via a home-delivered kit and a saliva sample, and some clinical guidance for those with positive test results. We believe that testing has substantial benefits, and therefore, we would like to expand testing to as many eligible women as possible.

**Abstract:**

There is limited information of the outcomes of direct-to-consumer testing for BRCA1 and BRCA2 mutations. The Screen Project was initiated in 2017 to offer BRCA1 and BRCA2 genetic screening to all Canadians over the age of 18 who wish to know their mutation status. Patients enrolled in the study from 2017 to 2019 and were followed for one year after the receipt of a genetic test result. Study subjects registered online and were sent a saliva sample kit, which was shipped to the reference laboratory. Pre-test genetic counselling and counselling for mutation-negative subjects was optional and at the individual’s discretion. There were 1269 tested individuals between March 2017 and January 2019. A total of 1157 (93%) were women and 87 (7%) were men. Sixty-six percent had a first- or second-degree relative with breast or ovarian cancer. Of the 1269 tested individuals, 30 (2.4%) had a pathogenic mutation in BRCA1 or BRCA2 (20 women and 10 men). Seventy-five percent of the female mutation carriers underwent a bilateral mastectomy and/or salpingo-oophorectomy within a year of receiving a positive result. Genetic counselling was available at no cost to all participants but was requested by only 5% of the non-carriers. The study subjects expressed a high degree of satisfaction with the process.

## 1. Introduction

Recently, there has been much discussion about widening the recommendations for genetic testing for breast cancer susceptibility [1,2,3,4,5,6]. The American College of Surgeons recommends that all women with breast cancer be tested [1]. The US Preventive Services Task Force extends this recommendation to include women who had breast cancer in the past [2,3]. It is generally agreed that all women with ovarian cancer should be tested [7]. Jewish women qualify by virtue of race alone—mutations are more frequent in Jewish women than in non-Jewish women—although the highest rates reported are in Bahamian women [8]. Other groups with a high prevalence of mutations include Trinidadian, Colombian and Mexican women [9,10].

There are three main reasons to get tested. First is to determine the future risk of cancer of the breast and ovary in order that appropriate preventive measures can be taken. Second is to manage a recently diagnosed breast cancer. Third is to offer at-risk relatives the opportunity to get tested themselves.

There has been little support to date for offering genetic testing to members of the general public regardless of race, sex, personal or family history of cancer [11,12]. The argument against universal access to testing is largely based on resources and cost, but concerns have also been raised about false positives, heightened anxiety and insurance denial. The potential benefits of a direct-to-consumer approach are to increase access to testing in individuals in remote areas, to remove the requirement for in-person referral, to reduce the burden of triage by the counsellor and the third-party payer and to reduce the cost per mutation detected. As our knowledge of cancer prevention and personalized treatment improves, we see mounting evidence in favor of open access testing. Between November 2017 and March 2019, we offered direct-to-consumer genetic testing for BRCA1 and BRCA2 to all Canadians over the age of 18 on a user-pay basis ($165 USD per test). We asked if the mutation yield is sufficiently high to justify testing and if patient satisfaction is high. We asked if the population of subjects who wish to be tested is representative of the Canadian population at large or is composed of men and women at high risk of carrying a mutation. We also asked the participants if there were ways in which we might improve the testing program.

## 2. Materials and Methods

### 2.1. Study Subjects

All Canadian men and women aged 18 years and older were eligible to participate. Enrollment was based on self-referral. There was no paid advertising for the Screen Project, but it was promoted through emails to physicians and genetic counsellors, the Canadian Cancer Society, patient advocacy groups and on the Women’s College Hospital (WCH) website. The project was also a featured topic in two Ontario newspapers. Interested participants were directed to a webpage developed specifically for the study on the WCH website. Potential participants were invited to watch a short pre-test counselling video, featuring a Women’s College Research Institute (WCRI) genetic counsellor. The video covers basic genetic concepts, types of test results and the implications of genetic testing. All subjects gave their informed consent for inclusion before they participated in the study. The study was conducted in accordance with the Declaration of Helsinki, and the protocol was approved by the WCH Research Ethics Board (2016-0112-B). Participants who wish to be tested were asked to provide informed consent and information on personal and family history of cancer. If potential participants wished further information, they were invited to contact the WCRI genetic counsellor by email or by telephone.

After completion of the first component, participants were directed to the Veritas Genetics study webpage. Participants were required to pay $165 USD to Veritas Genetics directly for covering the cost of a saliva collection kit, the shipping cost of the kit and the cost of the comprehensive screening of the BRCA1/2 genes. Veritas Genetics shipped a saliva collection kit to the participant. Samples were shipped by the participants directly to Veritas Genetics. Genetic testing was done at Veritas Genetics, a CLIA certified laboratory. Only the BRCA1 and BRCA2 genes were included in this test. Both small nucleotide changes and large rearrangements were screened for using Next Generation Sequencing technology and fragment analysis.

### 2.2. Results Disclosure

The genetic test result was released by Veritas Genetics to the WCRI study team. This result was reported as either: (1) negative, no mutation or a variant of uncertain significance (VUS) or (2) positive, a mutation with clinical impact was found.

Patients who received a positive result were contacted by the WCRI genetic counsellor and sent a copy of the genetic test result. All were offered a follow-up appointment, in person or by telephone, to discuss the implications of the result and to provide genetic counselling and to arrange appropriate referrals. Alternatively, they were referred to a cancer genetics counsellor in their home province.

Participants who received a negative result or a VUS (were notified by Veritas Genetics and a copy of the result was sent to the study team, but no routine follow-up was provided. Genetic counselling was available at the patient’s request, either by a genetic counsellor at Veritas Genetics or by WCRI genetic counsellor. Individuals with a reclassified VUS in future will be informed by Veritas according to Veritas Genetics’ policy.

### 2.3. Follow-Up

Participants were asked to complete a brief follow-up questionnaire after genetic testing designed to evaluate participants’ level of satisfaction. One year after the result disclosure, women with a positive result were asked about preventive measures taken, including preventive surgeries, MRI screening and tamoxifen.

## 3. Results

### 3.1. Who Was Tested?

Overall, we tested 1269 individuals between March 2017 and January 2019 (Table 1). Of these, 196 (15.4%) had a personal history of breast cancer, 6 (0.5%) had ovarian cancer, 10 (0.8%) had prostate cancer, 63 (5.0%) had another cancer and 994 (78.3%) had no cancer. The majority, 1157 (93%), were women and 87 (7%) were men. Two-thirds (66%) had a first- or second-degree relative with breast or ovarian cancer. All provinces were represented, but the majority of the participants (72%) were from Ontario. Also, 1117 (88%) had post-secondary or higher education.

The most common ways that participants learned about The Screen Project was through social media and news outlets (Table 2). Other important referral sources were friends and family members, and health care professionals (genetic counsellors, family doctors or specialist doctors).

### 3.2. Who Was Positive?

There were 30 individuals identified with a BRCA mutation, 14 in BRCA1 and 16 in BRCA2 (Table 3). This represents 2.4% of the tested individuals. Ten (33%) were men and 20 were women (67%). Of the 87 men tested, 10 (11.5%) were positive, compared to 20 of 1157 women (1.7%). Six of the BRCA carriers had a history of breast cancer (3.4% of the breast cancer patients who were tested). Seventeen (57%) had a first-degree relative with breast cancer and four (13%) had a first-degree relative with ovarian cancer. All 30 BRCA mutation carriers met NCCN guidelines for BRCA genetic testing, whereas 23 of the carriers met Ontario Ministry of Health criteria for the BRCA testing. There were also 51 individuals (4.0%) with a VUS detected.

### 3.3. Pedigree Examples

The clinical presentations and family histories of the 30 individuals with mutations varied widely. Eight mutation carriers (26.7%) had been referred previously for genetic testing and were declined testing under provincially-covered programs. For example, one woman was referred to a genetics clinic because her mother and aunt had ovarian cancer, but she was denied testing because their cancers were not pathologically confirmed (both women had died) (Figure 1A). She was found to be a BRCA1 mutation carrier.

One woman had a sister with pancreatic cancer at age 40 and a mother with breast cancer at age 40. She was denied testing because she did not meet local testing criteria. She was found to have a mutation in BRCA2 (Figure 1B).

One woman had requested testing because her mother had breast cancer but was denied. Several years later her identical twin sister was diagnosed with ovarian cancer. At this time, she became eligible for testing locally, but she opted for the Screen Project instead (Figure 1C). She had a mutation in BRCA2.

### 3.4. Patient Satisfaction

After completion of the study we sent a brief questionnaire by email to 400 women with a negative test result and 30 men and women with a positive test result. We asked if they were satisfied with the testing process and if they had any specific comments. We received a response from 18 of 30 (60%) carriers, and 141 of 400 (35.3%) non-carriers. Of the women with a mutation, 94% were satisfied and would recommend The Screen Project to a friend or a family member. Of the women without a mutation, 89% were satisfied and would recommend The Screen Project.

We received written comments from 129 participants about how we could improve the process. The most common recommendation was to lower the cost of the test, followed by a faster response time. Some offered the opinion that we could improve our visibility and marketing to include more patients. Other responses included expanding the number of genes tested, having additional resources for counseling and better communication of test results.

### 3.5. Preventive Measures Taken by Mutation Carriers

We contacted the female mutation carriers a minimum of one year after receipt of the test result. We asked if they had elected for any preventive measure to reduce their risk of breast and/or ovarian cancer. Of the 20 female carriers, we were able to contact 17 (85%) of them. Six of the women had a previous diagnosis of breast cancer at the time of testing. During the follow-up time, one woman died of breast cancer. Of the five living women, four women (80%) had a bilateral mastectomy, and one woman was scheduled for bilateral mastectomy. All five of the living breast cancer patients elected for bilateral salpingo-oophorectomy.

Eleven women were unaffected with breast cancer at the time of genetic testing. Within the follow-up period, two women were diagnosed with breast cancer (both were treated with bilateral mastectomy and bilateral salpingo-oophorectomy). Of the nine unaffected women, three women (33.3%) had bilateral mastectomy, two women were scheduled for bilateral mastectomy (22.2%), and four women (44.4%) were having MRI breast screening. Five of the nine unaffected women (55.6%) had a bilateral salpingo-oophorectomy. In summary, 75% (12/16) of the female BRCA mutation carriers had a bilateral mastectomy and/or with salpingo-oophorectomy.

## 4. Discussion

Based on our experience with The Screen Project, we support a model of health care delivery whereby genetic testing for BRCA1 and BRCA2 is made available to all Canadian adults at a low cost and at their own volition. This service is intended to complement, rather than to replace the existing provincially-funded genetics testing programs. Through guided direct-to-consumer testing, we seek to increase access to genetic testing and to remove systemic barriers, such as mandatory physician referral, long wait times and restrictive triage criteria. In many provinces, the absence of a living affected relative who is willing to be tested is an impediment to testing.

Our position differs from that of the U.S. Preventive Services Task Force in several respects. We do not believe that a primary health care provider is necessary to be the gate-keeper, i.e., that the primary care physician needs to conduct a preliminary risk assessment to see if a referral to a genetic counsellor is warranted. We think in-person pre-test genetic counselling should be available but should not be mandatory. It has been shown previously that the patient satisfaction was high among individuals who received pre-test counseling information by written and digital communication [13]. In the Screen Project, genetic counselling was available to all subjects at no additional cost; all women who tested positive spoke with a counsellor within six months of receiving the test result. Of those with a negative result, only 5% contacted the genetic counselor. Individuals who contacted the genetic counsellor were not significantly different from those who did not in terms of cancer risk, education or having a VUS.

The policy that testing be accompanied by genetic counselling was developed in the mid-1990s. In the conventional view, the purpose of pretest counselling is to inform the patient about the risks of breast (and ovarian) cancer, to discuss the risks and benefits of testing and to triage the patient for eligibility. With expanded testing, it is impractical to offer one-on-one counselling to all women. Consequently, many in the genetics community recommend that testing be offered only to high-risk women; the alternative is to make pre-test counselling optional. At 3000 dollars cost per test in the 1990s, it would be inefficient to test 50 women for one mutation (150,000 dollars per mutation found). However, in 2020, it is not reasonable to conduct 50 physician appraisals and genetic counselling appointments to find one mutation considering the current low cost of the test. We found 30 positive tests in 1269 subjects for a prevalence of 2.4%—by any account a reasonable yield—at an average cost per mutation detection of 7000–10,000 dollars. In theory, population testing is based on a lower expected prevalence of mutations than was seen here, but those who self-select to enter a genetic testing program are enriched for high-risk subjects. In our study, 66% of the enrollees had a first or second-degree relative with breast or ovarian cancer.

The U.S. Preventive Services Task Force recommended against testing to all women [2]. It is not clear how they based this decision. They allude to the “harms of risk assessment” without explicitly saying what these are. The Task Force implies that risk-reducing interventions are not helpful for women with a BRCA1 mutation but no family history of cancer, but to our knowledge there is no evidence that the benefit of interventions for BRCA1 or BRCA2 carriers with bona fide mutations are modified by the family history.

In our experience, men are often reluctant to pursue testing. In our study, only 7% of the participants were men, but surprisingly 12% of males were mutation carriers. In some cases, a female relative or spouse may encourage the male family members to get tested. In one family, two brothers were hesitant to get tested through conventional channels but agreed to send saliva specimens after the mother shipped kits to her sons. Both were found to carry a mutation in BRCA1 (Figure 1D).

In our study, 7 of 30 women (23%) who had a mutation did not meet provincial guidelines for testing and some had been denied testing. In a recent paper from Beitsch et al., 959 breast cancer patients were given genetic testing at their host institutions [14]. Of patients who had a mutation, 48% did not meet criteria from NCCN testing guidelines.

In this study, 75% of unaffected women with a mutation underwent bilateral mastectomy or salpingo-oophorectomy to reduce their cancer risks. This may be an indication that women who are willing to pay for genetic testing through the internet may also be motivated to undergo preventive surgery. The minority of women relied on MRI screening as the preventive strategy. MRI screening has been shown to be effective in down-staging breast cancer, but there is little data to date comparing breast cancer mortality in women who undergo MRI compared to those who have bilateral mastectomy or who are followed with mammography alone.

To date, few others advocate for open access genetic testing [11,12]. Presumably the main reason against universal testing is that, once recommended, the government will be obliged to pay for it—a valid concern but one that does not apply outside of the USA. In Canada, the government is not required to pay for preventive services if they are deemed useful by a consulting body and funding decisions are made at the provincial level. There are many other health services, such as IVF and screening colonoscopy, which are available privately and are not covered under public health schemes. In most of the developing world, the cost of counselling, testing and physician consultation is borne by the patient and the financial burden may be prohibitive [15]. We must be cautious in basing policy from the American Preventive Services Task Force to less-wealthy countries.

In many respects, our program was a success. The testing process was efficient, and the information was delivered in a timely manner. The yield of 2.4% and the cost per mutation identified were reasonable. The high yield of mutations indicates that the individuals who sought testing were not representative of the population at large and the yield cannot be used to estimate the prevalence of mutations in Canada. In addition, the subjects who sought testing were self-selected for high educational attainment (and possibly income).

The women with mutations had genetic counselling and the clinical course was the same as for women identified through conventional means. The patients reported a high degree of satisfaction. Nevertheless, the number of tested subjects was small, given the population of Canada. It is not clear why the participation rate was low; it may be because few women were aware of the Screen Project or they were aware but chose not to participate. We did not ask detailed questions about the impact of testing on patient knowledge or emotional distress and these areas will be explored in upcoming studies of an expanded cohort and will be compared with patients seeking testing through conventional channels.

There were no funds available for marketing or advertising. It was our hope that if a woman who met with a counselor and did not qualify for provincial testing, she would be referred to the Screen Project website. Future efforts will be focused on finding ways to increase uptake at a national level.

We would also like to target specific groups that were not well-represented to date, including women with DCIS [16] and men with prostate cancer [17,18]. We consider it to be inappropriate to treat a woman with DCIS or a man with prostate cancer expectantly without first ruling out a BRCA1 or BRCA2 mutation. There is also some evidence that knowledge of BRCA1 or BRCA2 mutation may direct personalized treatment of pancreatic cancer [19].

The mean age of testing of the women was 48-years old. The majority of breast cancers and approximately one-half of ovarian cancers in mutation carriers will occur before age 48 and ideally testing should occur between ages 25 and 30 [20,21]. The tested women were mostly well-educated. We did not ask about income (because of privacy issues) and it is possible that for some women, the lack of resources was an impediment to testing. Similar patterns of testing uptake have been seen in populations where testing is covered by insurance [22]. Although most tested individuals were white, all major ethnic groups were represented, including French-Canadians, East Asian, South Asian, indigenous, African and Hispanic among tested individuals.

We tested for only BRCA1 and BRCA2 in the phase 1 of the project, but we are revising our policy to include other cancer susceptibility genes as well. Phase 1 of the project reported here was planned in 2016 and ran in 2017 and 2018. Since then, our knowledge of other cancer susceptibility genes’ role has been increased continuously. We have now undertaken phase 2 of the project (stated in 2020). We evaluate 45 breast and other cancers susceptibility genes in this phase, and we include and classify patients with mutations and VUS. Patients have the option of having two BRCA1 and BRCA2 genes tested initially and if they are negative, they can have additional 43 genes tested for them. The concept of proper counselling has evolved over the past two decades and we will evaluate patient knowledge and satisfaction on an ongoing basis.

## 5. Conclusions

In conclusion, we endorse a model where genetic testing is offered direct to consumer via a home-delivered kit and a saliva sample, and some clinical guidance for those with positive test results. Genetic counselling and physician referral are optional prior to testing, but not mandatory. Those who test positive should be counselled and offered a personal referral to genetic counselors and other health professionals in their area for follow-up and testing family members. We believe that testing has substantial benefits and therefore we would like to expand testing to as many eligible women as possible.

## Figures and Tables

**Figure 1 cancers-13-01894-f001:**
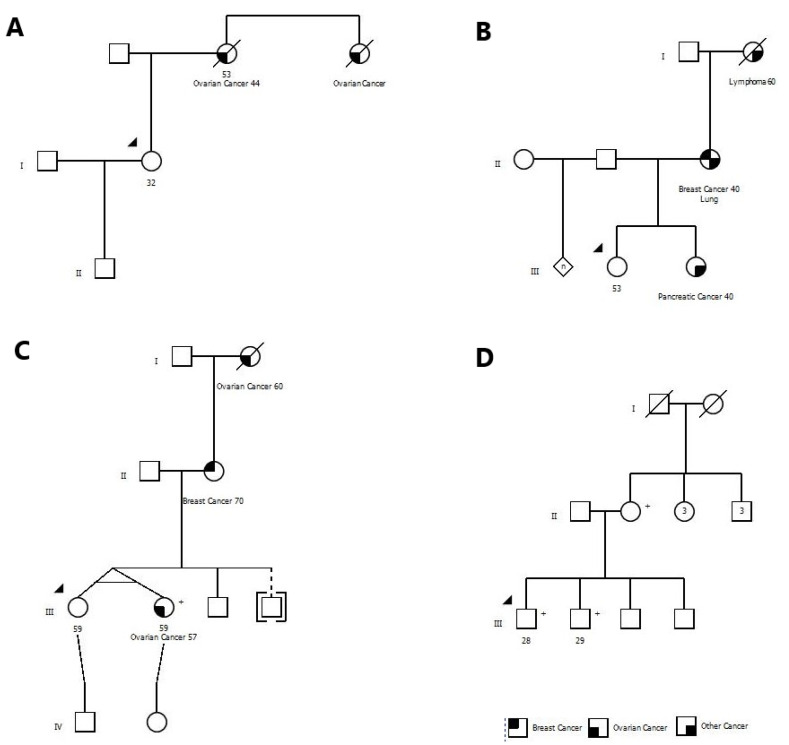
The pedigrees for five BRCA mutation carriers have been shown. (**A**) Proband was denied testing because of not having pathological confirmation of cancers in her family members. (**B**) Proband was denied testing based on local testing criteria. (**C**) Proband was qualified for testing under healthcare system, however, she decided to do the testing through the Screen Project because of ease of access. (**D**) Testing was ordered by the mother who was a known BRCA mutation carrier for her two adult sons, and both were positive. (+) sign denotes mutation carriers.

**Table 1 cancers-13-01894-t001:** Characteristics of the 1269 study participants.

Variables	Mean (Range) and Frequency (Percent)
Mean Age	47.9 (19–87)
Age Group	
18–25	18 (1.5)
25–35	188 (15.1)
35–50	521 (41.9)
50–60	320 (25.7)
60+	197 (15.8)
Missing	25
Sex	
Female	1157 (93.0)
Male	87 (7.0)
Missing	25
Province	
Ontario	892 (72.2)
British Columbia	117 (9.5)
Alberta	57 (4.6)
Saskatchewan	42 (3.4)
Quebec	40 (3.2)
Manitoba	31 (2.5)
Nova Scotia	28 (2.3)
New Brunswick	14 (1.1)
Newfoundland	8 (0.7)
Prince Edward Island	3 (0.2)
Yukon	2 (0.2)
Nunavut	1 (0.1)
Missing	34
Ethnic group *	
Jewish	89 (7.0)
French–Canadian	118 (9.3)
Other white	1137 (89.6)
Asian	82 (6.5)
Aboriginal/indigenous	25 (2.0)
African-American	12 (0.9)
Other	162 (12.8)
Missing	24
Education	
Elementary/other	14 (1.1)
High school	114 (9.2)
Post-secondary	729 (57.4)
Graduate school	378 (29.8)
Missing	34
Previous cancer	
None	994 (78.5)
Breast/DCIS	196 (15.4)
Ovarian	6 (0.5)
Prostate	10 (0.8)
Pancreas	0 (0.0)
Other	63 (5.0)

* Each participant can have more than one ethnic group.

**Table 2 cancers-13-01894-t002:** Sources of referral to the Screen Project.

Source	Number (%)
Social media (Facebook, Twitter)	215 (16.9)
Friend	172 (13.6)
Genetic counsellor	127 (10.0)
Internet	120 (9.5)
Family doctor	120 (9.5)
Magazine/newspaper	107 (8.4)
Specialist physician	59 (4.6)
Television	55 (4.3)
Family member	51 (4.0)
Women’s College Hospital	45 (3.5)
Canadian Cancer Society	4 (0.3)
Other	100 (7.9)
Missing	94

**Table 3 cancers-13-01894-t003:** Individuals with mutations identified in the Screen Project.

Mutated Gene	Age	Sex	Province	History of Cancer	Number of First or Second Degree Relatives with Cancer		
					Breast	Ovary	Other
BRCA1	44	M	Ontario	Leukemia	2	0	1
BRCA1	38	F	Quebec		1	1	0
BRCA2	44	F	Alberta		1	1	0
BRCA1	56	F	Ontario	Breast	0	0	1
BRCA1	32	F	New Brunswick		0	2	0
BRCA2	53	F	Ontario		1	0	2
BRCA2	46	F	British Columbia	Breast	2	0	4
BRCA1	43	F	Ontario		1	0	1
BRCA2	38	M	Ontario		5	0	2
BRCA1	36	F	British Columbia		1	0	1
BRCA2	38	F	Ontario		1	0	3
BRCA1	40	M	Ontario		3	0	0
BRCA2	32	M	Ontario		1	0	0
BRCA2	40	F	Nova Scotia		1	0	3
BRCA2	35	M	Ontario		2	0	0
BRCA2	59	F	Alberta		1	2	0
BRCA2	45	M	Quebec		1	0	0
BRCA2	51	F	Ontario	Breast	1	0	2
BRCA1	28	M	Ontario		0	0	0
BRCA2	42	F	Ontario		2	0	0
BRCA1	29	M	Ontario		0	0	0
BRCA1	52	F	Ontario		1	0	0
BRCA2	30	F	Newfoundland	Breast	2	0	2
BRCA2	44	F	Ontario		2	0	1
BRCA1	52	F	Ontario	Breast	0	0	2
BRCA2	57	F	Quebec	Breast	1	1	0
BRCA1	19	F	Ontario		2	0	0
BRCA2	44	M	Ontario		2	1	2
BRCA1	64	M	Ontario		1	0	0
BRCA1	28	F	Ontario		2	1	0

## Data Availability

Not applicable.

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
