# Peer review of "The Screen Project: Guided Direct-To-Consumer Genetic Testing for Breast Cancer Susceptibility in Canada"

_cancers, 2021, doi:10.3390/cancers13081894_

Round 1

Reviewer 1 Report

The authors have adequately addressed the concerns from the initial review.

Author Response

Thank you!

Reviewer 2 Report

Prior to 2020 the NCCN guidelines focused largely on testing criteria for BRCA1/2 and appropriate risk management for carriers of a BRCA1 or BRCA2 pathogenic or likely pathogenic variant. Given the strong evidence that genes beyond BRCA1/2 markedly increased risk of breast and/or ovarian cancers, the testing criteria has been expanded to High-Penetrance Breast and/or Ovarian Cancer Susceptibility Genes. The criteria have been reorganized into 3 sections: (1) testing is clinically indicated, (2) testing may be considered, and (3) low probability of testing results having documented clinical utility. 

In line 273 the authors state that “in many respect our program was a success”. However, in order to properly evaluate that the reader need to know how many of the negative/ VUS for BRCA1/2 were among the three groups. The authors only included the patients who were positive. This is important since negative and VUS subjects will not receive genetic counseling even if they are at high-risk for other cancer genes and as the authors indicated that there is there is no plan to test patients from phase 1 who meet NCCN guideline for additional genes other than BRCA1 and BRCA2.

This is a major limitation of the program at current form.  There should be a plan for patients who are negative/VUS for BRCA1/2 BUT meet the NCCN criteria for testing to receive proper counseling or recommendation for referral to counseling otherwise they will be falsely reassured of their cancer risk.

Author Response

The goal of the Screen Project was to offer genetic testing to everybody regardless of any criteria. Our goal was to identify mutation carriers through this program who do not have access to testing or who do not meet the criteria recommended by NCCN or other organizations.

In this first phase of the project our criteria for success was identification of mutations in those genes that were included (BRCA1 and BRCA2)  and this is the subject of the first report. This phase 1 of the project was planned in 2016 and ran in 2017 and 2018.

Since then, our knowledge of other cancer susceptibility genes' role has been increased continuously as reflected in the current NCCN guideline.

We have now undertaken Phase 2 of the project (started in 2020). We evaluate 43 breast (and other cancer) susceptibility genes and we include and classify patients with mutations and VUS in this study.

Patient will have the option of having two genes tested initially and if they are negative, they can have additional 43 genes tested and the two groups will be compared.

The concept of proper counselling has evolved over the past two decades and we will evaluate patient knowledge and satisfaction on an ongoing basis.

Round 2

Reviewer 2 Report

The information included in the response to reviewer in particular the expansion of the future testing to include 43 genes is not clear in the discussion in the current form (there is only description of PALB2 and CHEK2 testing) so please expand the discussion to reflect that. 

There should be a mechanism for subject enrolled in phase 1 of the project who have strong personal or family history of cancer but negative BRCA1 / BRCA2 testing to be eligible for the expanded genetic testing in the 2nd phase of the project.

Please include the following language from the response to reviewer in the discussion.  

" In this first phase of the project our criteria for success was identification of mutations in those genes that were included (BRCA1 and BRCA2) and this is the subject of the first report. This phase 1 of the project was planned in 2016 and ran in 2017 and 2018.

Since then, our knowledge of other cancer susceptibility genes' role has been increased continuously as reflected in the current NCCN guideline.

We have now undertaken Phase 2 of the project (started in 2020). We evaluate 43 breast (and other cancer) susceptibility genes and we include and classify patients with mutations and VUS in this study.

Patient will have the option of having two genes tested initially and if they are negative, they can have additional 43 genes tested and the two groups will be compared.

The concept of proper counselling has evolved over the past two decades and we will evaluate patient knowledge and satisfaction on an ongoing basis."

Author Response

We added the information about testing additional 43 genes in the second phase of the project in the lines 301-310.

This manuscript is a resubmission of an earlier submission. The following is a list of the peer review reports and author responses from that submission.

Round 1

Reviewer 1 Report

The authors present their pilot work of a direct-to-consumer genetic testing model for breast cancer susceptibility to address some of the challenges with conventional provider-based testing. The manuscript is well written but has several limitations that need to be addressed.

  • The authors combined cases with negative testing and VUS. How many VUS were identified? It is not clear if patient with VUS received any genetic counseling and the plan to contact them if the variant is reclassified.
  • Only 5% of subjects with negative/VUS contacted genetic counselor, please expand on these subjects (cancer risk, education, negative vs VUS..etc).
  • Only two genes were tested, the authors indicate that they are planning to include PALB2. The statement about CHEK2 is not clear if they plan to include that or not. How about other breast cancer genes? Is there any plan to retest NCCN high risk patients for additional cancer genes?
  • The authors state that “In many respects, our program was a success” but in order to objectively estimate that the authors also need to present the percent of NCCN estimated high-risk patients missed using the proposed strategy. Since NCCN high-risk patients who are negative or with VUS don’t routinely receive counseling, how to insure that they receive proper follow up and not falsely reassured? These limitations need to be properly discussed.
  • How well the study represent different ethnicities in Canada? Were any ethnic group missed? Were there any language barriers? please discuss.
  • Is this model more appropriate for patients with certain level of education/resources? please discuss.

Author Response

The authors combined cases with negative testing and VUS. How many VUS were identified?

We added the info about the number of detected VUS in the manuscript lines 133 and 134.

We have stated (on line 100) that those with negative results did not receive active genetic counselling.    However, we clarified this to include those with VUS as well.

It is not clear if patient with VUS received any genetic counseling and the plan to contact them if the variant is reclassified.

We have added a sentence on lines 103 and 104 about informing patients with a reclassified VUS in future.

Only 5% of subjects with negative/VUS contacted genetic counselor, please expand on these subjects (cancer risk, education, negative vs VUS..etc).

It was explained on lines 213-215 that the 5% of individuals with negative results who requested genetic counselling were not different from the rest in terms of their risk of cancer, education or VUS.

 Only two genes were tested, the authors indicate that they are planning to include PALB2. The statement about CHEK2 is not clear if they plan to include that or not. How about other breast cancer genes? Is there any plan to retest NCCN high risk patients for additional cancer genes?

Phase 2 of the Screen Project was launched in June 2020 with the option of requesting analysis of additional 43 cancer susceptibility genes (including PALB2 and CHEK2 and 41 other genes) other than BRCA1 and BRCA2 genes with no extra charge. We hope to report the initial results of the second phase in 2022.

The authors state that “In many respects, our program was a success” but in order to objectively estimate that the authors also need to present the percent of NCCN estimated high-risk patients missed using the proposed strategy. Since NCCN high-risk patients who are negative or with VUS don’t routinely receive counseling, how to insure that they receive proper follow up and not falsely reassured? These limitations need to be properly discussed.

There is no plan to test patients from phase 1 who meet NCCN guideline for additional genes other than BRCA1 and BRCA2.

The proposed strategy is universal population-based testing for BRCA genes and with that strategy, there will be no missed carriers. We are not sure what the reviewer means from the percent of NCCN estimated high-risk patients that might be missed using the proposed strategy.

How well the study represent different ethnicities in Canada? Were any ethnic group missed? Were there any language barriers? please discuss.

We discussed the ethnicity and language issue on lines 303-308.

Is this model more appropriate for patients with certain level of education/resources? please discuss.

The population-based model is good for everybody regardless of their education and socioeconomic status.

Reviewer 2 Report

The manuscript by Narod et al is an important and interesting study on population based screening of BRCA1 and BRCA2 mutations in Canada.  Overall, it is of high quality and significance.  I have only a few comments.

First, the layout and resolution of Figure 1 needs improvement.  There is too much white space and there are some unexplained features (e.g. brackets around a male sibling in 1C and the number 3 inside a male and female in 1D).  Also, the Figure legend is too brief and lacks detail.

There are two minor typos:

Line 210 'one-on-one'

Line 267 'and possibly'

Otherwise, I find this paper of high quality and worthy of publication.

Author Response

We appreciate the positive feedback received from the reviewer.

We fixed the layout of figure 1, and we are also submitting a high-quality image separately to be used.

We will also put a complete description of the pedigrees shown in figure 1

The pedigree symbols used are standard symbols. The male with the bracket on figure 1C means that that person was adopted in. The numbers inside the individual symbol also mean that there are that many number siblings of that sex.

The typos were also fixed.